# From Relevance to Utility: Evidence Retrieval with Feedback for Fact Verification

**Hengran Zhang**[1,2]  **Ruqing Zhang**[1,2] *  **Jiafeng Guo**[1,2]  **Maarten de Rijke**[3]
**Yixing Fan**[1,2]  **Xueqi Cheng**[1,2]

[1]School of Computer Science and Technology, University of Chinese Academy of Sciences;
[2]CAS Key Laboratory of Network Data Science and Technology,
Institute of Computing Technology, Chinese Academy of Sciences; [3]University of Amsterdam.
{zhanghengran22z, zhangruqing, guojiafeng, fanyixing, cxq}@ict.ac.cn, m.derijke@uva.nl

## Abstract

Retrieval-enhanced methods have become a primary approach in fact verification (FV); it requires reasoning over multiple retrieved pieces of evidence to verify the integrity of a claim. To retrieve evidence, existing work often employs off-the-shelf retrieval models whose design is based on the probability ranking principle. We argue that, rather than relevance, for FV we need to focus on the utility that a claim verifier derives from the retrieved evidence. We introduce the *feedback-based evidence retriever* (FER) that optimizes the evidence retrieval process by incorporating feedback from the claim verifier. As a feedback signal we use the divergence in utility between how effectively the verifier utilizes the retrieved evidence and the ground-truth evidence to produce the final claim label. Empirical studies demonstrate the superiority of FER over prevailing baselines.[1]

## 1 Introduction

The risk of misinformation has increased the demand for fact-checking, i.e., automatically assessing the truthfulness of textual claims using trustworthy corpora, e.g., Wikipedia. Existing work on fact verification (FV) commonly adopts a retrieval-enhanced verification framework: an evidence retriever is employed to query the background corpus for relevant sentences, to serve as evidence for the subsequent claim verifier. High-quality evidence is the foundation of claim verification. Currently, prevailing approaches to identifying high-quality evidence typically adopt off-the-shelf retrieval models from the information retrieval (IR) field for evidence retrieval (Wan et al., 2021; Jiang et al., 2021; Chen et al., 2022a; Liu et al., 2020). These models are usually based on the probability ranking principle (PRP) (Robertson, 1977), ranking sentences

based on their likelihood of being relevant to the claim. However, the sentences retrieved in this way are assumed to be consumed by humans, and this may not align with a retrieval-enhanced verification framework (Zamani et al., 2022): the top-ranked sentences produced by existing retrieval models do not always align with the judgments made by the claim verifier regarding what counts as evidence.

We argue that when designing evidence retrieval models for FV, the notion of relevance should be reconceptualized as the utility that the verifier derives from consuming the evidence provided by the retrieval model, a viewpoint that aligns well with task-based perspectives on IR (Kelly et al., 2013). Hence, we assume that *when training evidence retrievers, it is advantageous to obtain feedback from the claim verifier as a signal to optimize the retrieval process*.

Therefore, we propose a shift in emphasis from relevance to utility in evidence retrieval for FV. We introduce the *feedback-based evidence retriever* (FER) that incorporates a feedback mechanism from the verifier to enhance the retrieval process. FER leverages a coarse-to-fine strategy. Initially, it identifies a candidate set of relevant sentences to the given claim from the large-scale corpus. Subsequently, the evidence retriever is trained using feedback from the verifier, enabling a re-evaluation of evidence within the candidate set. Here, feedback is defined as the utility divergence observed when the verifier evaluates the sentences returned by the retriever, compared to when it consumes the ground-truth evidence for predicting the claim label. By measuring the utility criterion between the two scenarios, we can optimize the retriever specifically for claim verification.

Experimental results on the large-scale Fact Extraction and VERification (FEVER) dataset (Thorne et al., 2018) demonstrate a 23.7% F1 performance gain over the SOTA baseline.

---

*Research conducted when the author was at the University of Amsterdam.

[1]Our code and data are available at https://github.com/ict-bigdatalab/FER

## 2 Background

For the FEVER task (Thorne et al., 2018), automatic FV systems need to determine a three-way veracity label, i.e., SUPPORTS, REFUTES or NOT ENOUGH INFO, for each human-generated claim based on a set of supporting evidence from Wikipedia. The FV system can be split into document retrieval, evidence retrieval, and claim verification phases. The first phase is to retrieve related documents from the corpus. The second phase aims to extract evidence from all sentences of the recalled documents. The final phase aggregates the information of the retrieved evidence to predict a claim label. In this work, our contributions are focused on the evidence retrieval step. The related work and more extensive discussions are included in Appendix A.1.

## 3 Our Approach

Based on the related documents (i.e., Wikipedia pages) obtained by a common document retrieval method with entity linking (Hanselowski et al., 2018; Liu et al., 2020), FER comprises two components: (i) *coarse retrieval*: using a PRP-based retrieval model to recall a set of candidate sentences to the claim from the related documents; and (ii) *fine-grained retrieval*: selecting evidential sentences from the candidate set based on the feedback from the claim verifier. The overall architecture of FER is illustrated in Figure 1.

### 3.1 Coarse retrieval

For a given claim, we first retrieve a set $S$ of candidate sentences from the related documents based on the PRP. Following (Zhou et al., 2019; Liu et al., 2020), we employ the base version of BERT for coarse retrieval. We utilize the hidden state of the "[CLS]" token to represent the claim and sentence pair. To project the "[CLS]" hidden state to a ranking score, we employ a linear layer followed by a tanh activation function. Lastly, we optimize the BERT-based ranking model using a typical pairwise loss function. During inference, given a test claim, we take the top-$K$ ranked sentences to form the candidate set $S$.

### 3.2 Fine-grained retrieval with feedback

Given the candidate set $S$, we train the fine-grained evidence retriever $R_\theta$ with the feedback from the claim verifier. We leverage the base version of BERT to implement $R_\theta$. The claim and all sentences in $S$ are concatenated as a single input sequence, with a special token $[CLS]$ added at the

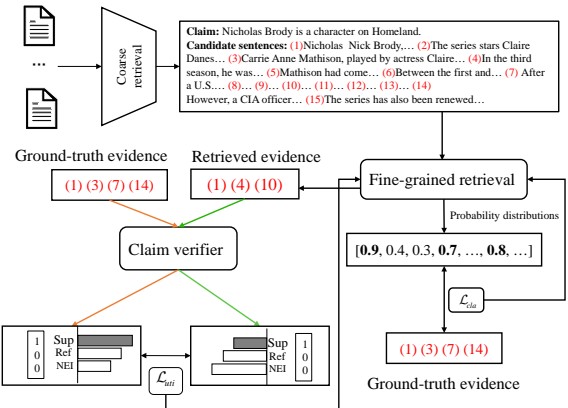

Figure 1: Architecture of FER.

beginning of each sentence to indicate its presence. The training objective $\mathcal{L}$ for $R_\theta$ consists of two parts, i.e., an *evidence classification* $\mathcal{L}_{cla}$ and a *utility divergence* $\mathcal{L}_{uti}$, denoted as:

$$\mathcal{L} = \alpha\mathcal{L}_{cla} + \beta\mathcal{L}_{uti},$$

where $\alpha$ and $\beta$ are coefficients.

**Evidence classification.** We first leverage the ground-truth evidence to boost plausibility. Specifically, we input the concatenated sequence into BERT and leverage the hidden state of each $[CLS]$ to classify the corresponding sentence through linear layers and ReLU activation functions. Here, $g \in \{0, 1\}$ is the sentence label where 1 or 0 represents whether a sentence is the ground-truth evidence or not. Then, the classification loss is defined as a measure of the difference between the predicted evidence and the ground-truth evidence via binary cross-entropy loss, i.e.,

$$\mathcal{L}_{cla} = -\sum_{s \in S} g\log(p(s)) + (1 - g)\log(1 - p(s)),$$

where $s$ is a sentence in the candidate set $S$ and $p(s)$ is the predicted probability of $s$ by $R_\theta$ as the ground-truth. During training, this loss function $\mathcal{L}_{cla}$ encourages the retriever to select evidence sentences that are deemed plausible, representing ground-truth evidence.

**Utility divergence based on the claim verifier.** To further ensure that the retrieved evidence improves verification performance, we propose to get feedback from the claim verifier as a signal for optimizing the evidence retriever. Here, we leverage the utility that the verifier obtains by consuming the retrieved evidence as the feedback. Specifically, for each claim $c$, we can measure the divergence in utility between the verifier's judgements based on the ground-truth evidence and those based on the

Table 1: Comparisons of the evidence retrieval performance achieved by FER and baselines. † indicates statistically significant improvements over all the baselines (p-value < 0.05).

| Model | Dev | | | Test | | |
|---|---|---|---|---|---|---|
| | P | R | F1 | P | R | F1 |
| TF-IDF (Thorne et al., 2018) | - | - | 17.20 | 11.28 | 47.87 | 18.26 |
| ColumbiaNLP (Chakrabarty et al., 2018) | - | 78.04 | - | 23.02 | 75.89 | 35.33 |
| UKP-Athene (Hanselowski et al., 2018) | - | 87.10 | - | 23.61 | 85.19 | 36.97 |
| GEAR (Zhou et al., 2019) | 24.08 | 86.72 | 37.69 | 23.51 | 84.66 | 36.80 |
| NSMN (Nie et al., 2019a) | 36.49 | 86.79 | 51.38 | 42.27 | 70.91 | 52.96 |
| KGAT (Liu et al., 2020) | 27.29 | **94.37** | 42.34 | 25.21 | **87.47** | 39.14 |
| DREAM (Zhong et al., 2020) | 26.67 | 87.64 | 40.90 | 25.63 | 85.57 | 39.45 |
| DQN (Wan et al., 2021) | 54.75 | 79.92 | 64.98 | 52.24 | 77.93 | 62.55 |
| GERE (Chen et al., 2022a) | 58.43 | 79.61 | 67.40 | 54.30 | 77.16 | 63.74 |
| Stammbach (Stammbach, 2021) | 71.25 | 83.21 | 76.72 | - | - | - |
| FER w/o $\mathcal{L}_{cla}$ | 80.26† | 81.62 | 80.93† | 75.76† | 75.97 | 75.86† |
| FER w/o $\mathcal{L}_{uti}$ | 75.69† | 87.38 | 81.12† | 71.84† | 80.87 | 76.08† |
| FER | **84.04**† | 84.59 | **84.31**† | **79.35**† | 78.34 | **78.84**† |

sentences provided by the retriever when predicting the claim label, i.e.,

$$\mathcal{L}_{uti} = y^* D_\phi(c, E^*) - y^* D_\phi(c, R_\theta(c, S)),$$

where $c$ is the given claim, $E^*$ denotes the ground-truth evidence and $y^*$ denotes a one-hot indicator vector of the ground-truth claim label. $R_\theta(c, S)$ represents the sentences selected by the evidence retriever, which takes the candidate set $S$ and the claim $c$ as input. $D_\phi(\cdot)$ is the probability distribution predicted by the claim verifier. The details of loss gradient passed to the fine-grained retriever can be found in the Appendix A.2.

Here, we also use BERT's base version as the verifier in FER. During training, the verifier performs classification on the concatenated input, i.e., [CLS] + $c$ + [SEP] + $E^*$ + [SEP], and leverage the hidden state of [CLS] to classify $c$ into three categories via a linear layer and softmax function. After training, the verifier is fixed and we directly compute the probability prediction vector $D_\phi(\cdot)$ based on $E^*$ and $R_\theta(c, S)$, respectively. Note that we can leverage other advanced claim verifiers to provide feedback (Section 4.4). The loss function $\mathcal{L}_{uti}$ encourages the retriever to prioritize the selection of sentences that are crucial for claim verification.

At inference time, given a test claim, we leverage the optimized retriever to retrieve evidence (Section 4.2). Then, we can directly input the retrieved evidence to several advanced verification models to verify the claim (Section 4.3).

## 4 Experiments

### 4.1 Experimental settings

We conduct experiments on the FEVER benchmark dataset (Thorne et al., 2018), which com-

prises 185,455 claims with 5,416,537 Wikipedia documents from the June 2017 Wikipedia dump. All claims are annotated as "SUPPORTS", "REFUTES" or "NOT ENOUGH INFO".

For evaluation, we leverage official evaluation metrics, i.e., Precision (P), Recall (R), and F1 for evidence retrieval, and the FEVER score and Label Accuracy (LA) for claim verification. We prioritize the F1 as our primary metric for evidence retrieval because it directly reflects the model performance in terms of retrieving precise evidence.

The candidate set size $K$ is set to 25; both $\alpha$ and $\beta$ are set to 1. All hyper-parameters are tuned on the development set. We include detailed descriptions of the implementation details, evaluation metrics, and baselines in Appendix A.3.

### 4.2 Results on evidence retrieval

We select several representative evidence retrieval models as our baselines and conduct experiments on both the development and test set following the common setup in FEVER. Based on the results presented in Table 1, we find that: (i) FER significantly outperforms the state-of-the-art methods in terms of P and F1, demonstrating its superiority in providing supporting evidence for claim verification based on utility. (ii) The superior precision achieved by FER comes at the expense of lower recall when compared to baselines. The reason might be that FER aims to retrieve more precise and concise sets of evidence for each claim. That is, FER results in a smaller amount of retrieved evidence compared to the baselines. Similarly, DQN, GERE and Stammbach also aim to identify precise evidence. However, their P and F1 scores are substantially lower than those for FER, indicating that

Table 2: Performance of different claim verification models on the test set using evidence from the original paper vs. evidence retrieved by FER.

| Model | LA | FEVER |
|---|---|---|
| BERT Concat (Zhou et al., 2019) | 71.01 | 65.64 |
| BERT Concat + FER | 72.46 | 68.16 |
| BERT Pair (Zhou et al., 2019) | 69.75 | 65.18 |
| BERT Pair + FER | 72.12 | 68.03 |
| GEAR (Zhou et al., 2019) | 71.60 | 67.10 |
| GEAR + FER | 72.54 | 67.49 |
| GAT (Liu et al., 2020) | 72.03 | 67.56 |
| GAT+FER | 72.85 | 69.43 |
| KGAT(BERT Base) (Liu et al., 2020) | 72.81 | 69.40 |
| KGAT+FER | 73.34 | 69.61 |

considering the utility of evidence contributes to retrieving evidential sentences for FV. (iii) FER without $\mathcal{L}_{cla}$ outperforms FER without $\mathcal{L}_{uti}$ in terms of P, indicating that feedback from the verifier plays a crucial role in aiding the retriever to identify precise evidence that is crucial for the verification process. FER without $\mathcal{L}_{cla}$ retrieves a smaller number of sentences than FER without $\mathcal{L}_{uti}$, i.e., its performance in terms of R and F1 is poorer. Some case studies can be found in Appendix A.4.2.

### 4.3 Results on claim verification

To better understand the effectiveness of evidence retrieved by FER, we choose several advanced claim verification models, and provide them with evidence retrieved by FER and evidence obtained from the original paper, respectively. The experiments are reported on the test set; see Table 2. Similar findings can be obtained on the development set; see Appendix A.4.1. All the claim verification models leveraging evidence retrieved by FER outperform their respective original versions based on PRP. By conducting further analyses, we find that the evidence retrieved based on the likelihood of being relevant to the claim in the original papers may not be always useful for the verification process or contain conflicting pieces. FER is able to select more precise evidence for claim verification, contributing to the verification outcome and leading to improved results.

Table 3: Evidence retrieval performance with feedback from different verifiers on the test set.

| Verifier | P | R | F1 |
|---|---|---|---|
| GAT | 77.93 | 79.48 | 78.70 |
| KGAT | 84.38 | 75.95 | 79.94 |

### 4.4 Quality analysis

**Feedback using different verifiers**. Here, we explore the potential of utilizing off-the-shelf claim verifiers to offer feedback to $R_\theta$ in our FER. From Table 2, KGAT (Liu et al., 2020) and GAT exhibit promising verification performance, making them suitable choices for providing feedback. BERT Pair cannot be employed for providing feedback to FER, since it produces multiple independent probabilities instead of a probability distribution encompassing the final claim labels. In future work, we will explore alternative feedback formulations to incorporate additional verifiers. The experiments are reported on the test set; see Table 3. Similar findings have been obtained on the development set; see Appendix A.4.1. Obtaining feedback from other verifiers could also show promising evidence retrieval performance; this result showcases the adaptability of the proposed FER method, which effectively adjusts to different verifiers. Besides, good retrievers and verification models can mutually benefit from each other's strengths.

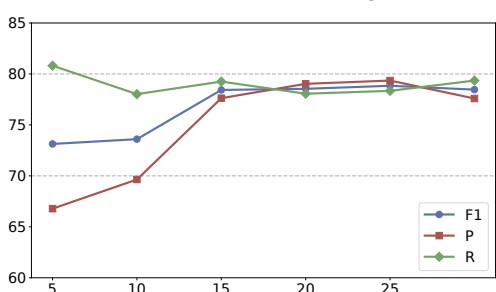

Figure 2: Evidence retrieval performance with different values of $K$ on the test set.

**Effect of the candidate set size $K$**. In Figure 2, we plot the effect of the number of candidate sentences $K$ obtained by the coarse retrieval components on the final evidence retrieval performance. Recall is high for all values of $K$. The reason might be that on average, each claim is associated with 1.86 relevant sentences in FEVER, while $K$ is set to be larger than 5. Furthermore, a lower value of $K$ implies that less information will be available for the fine-grained retrieval model, which, in turn, hurts precision. As $K$ is increased, more information is provided to the model, which amplifies the impact of feedback, potentially causing slight fluctuations in the recall rate.

### 5 Conclusion

In this paper, we proposed FER (*feedback-based evidence retrieval*) for FV, a novel evidence retrieval framework that incorporates feedback from the claim verifier. Unlike traditional approaches

based on PRP, FER places an emphasis on the utility of evidence that contributes meaningfully to the verification process, going beyond mere relevance. Through the integration of feedback from the verifier, FER effectively identifies the evidence that is both relevant and useful for claim verification. The experimental results on the FEVER dataset demonstrated the effectiveness of FER.

## Limitations

In this paper, we utilized the performance disparity between ground-truth evidence and retrieved evidence for claim verification as a form of feedback to train an evidence retrieval model. There are two primary limitations that should be acknowledged: (i) Currently, the retrieval and verification models are optimized independently, lacking conditional optimization or joint end-to-end optimization. Future research could explore approaches to optimize both the retrieval and verification models in a single objective function. In this manner, we expect to see improved overall performance. (ii) In this study, we focused solely on investigating the probability distribution generated by the claim verifier to compute the utility divergence. However, it is important to note that there exist multiple methods for quantifying the divergence in utility of retrieval results, e.g., the gradients of the verification loss. Additionally, it is worth considering that there may be various forms of feedback that can be incorporated. We hope that this research will inspire further exploration and attention in this area for future studies. (iii) Finally, we only demonstrated the effectiveness of the proposed FER method on a single dataset, the FEVER dataset, and the evidence retrieval process. We encourage future work aimed at the creation of further claim verification datasets and document retrieval process.

## Acknowledgments

This work was funded by the National Natural Science Foundation of China (NSFC) under Grants No. 62006218 and 61902381, the China Scholarship Council under Grants No. 202104910234, the Youth Innovation Promotion Association CAS under Grants No. 2021100, the project under Grants No. JCKY2022130C039 and 2021QY1701, and the Lenovo-CAS Joint Lab Youth Scientist Project. This work was also (partially) funded by the Hybrid Intelligence Center, a 10-year program funded by the Dutch Ministry of Education, Culture and Science through the Netherlands Organisation for Scientific Research, https://hybrid-intelligence-centre.nl, and project LESSEN with project number NWA.1389.20.183 of the research program NWA ORC 2020/21, which is (partly) financed by the Dutch Research Council (NWO).

All content represents the opinion of the authors, which is not necessarily shared or endorsed by their respective employers and/or sponsors. We would like to thank the reviewers for their valuable feedback and suggestions.

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

# A  Appendix

## A.1  Related work

In this section, we briefly review two lines of related work, i.e., fact verification and retrieval-enhanced machine learning.

### A.1.1  Fact Verification

The real-world fact verification (FV) task focuses on verifying the accuracy of claims made by humans through the retrieval of evidential sentences from reliable corpora such as Wikipedia (Chen et al., 2022a). Current FV models typically adopt a three-step pipeline framework, which includes document retrieval, evidence retrieval, and claim verification.

In the document retrieval stage, existing works could be broadly categorized into three categories: entity linking (Hanselowski et al., 2018; Chernyavskiy and Ilvovsky, 2019; Zhao et al., 2020; Liu et al., 2020), keyword matching (Nie et al., 2019a; Ma et al., 2019; Nie et al., 2019b) and feature-based (Hidey and Diab, 2018; Yang et al., 2017; Chen et al., 2022a) methods.

In the evidence retrieval stage, existing FV works commonly utilize retrieval models that are built upon the probability ranking principle (PRP) (Robertson, 1977) in the field of information retrieval (IR). These models aim to select the top-ranked sentences as evidence based on relevance ranking (Jiang et al., 2021; Soleimani et al., 2020; Zhou et al., 2019; Liu et al., 2020; Hanselowski et al., 2018; Wan et al., 2021). Very recently, Chen et al. (2022a); Liu et al. (2023) proposed to retrieve evidence in a generative fashion by generating document titles and evidence sentence identifiers.

In the claim verification stage, considerable attention has been given to this step in many existing FV works. These mainly include implication-based inference methods that treat the verification task as an entailment task (Thorne et al., 2018; Nie et al., 2019a; Krishna et al., 2022; Nie et al., 2019b), as well as graph neural network-based methods (Liu et al., 2020; Zhou et al., 2019) that frame the verification task as a graph reasoning task. It is worth noting that the accuracy of this stage heavily relies on the retrieved evidence. Therefore, having a precise set of evidence can significantly impact the verification outcome and lead to improved results.

### A.1.2  Retrieval-enhanced machine learning

As mentioned in Zamani et al. (2022), one approach to improve accuracy is by increasing the parameter size of machine learning models. However, it is important to consider that this approach comes with the trade-off of increased costs in model design and training. Hence, the task of alleviating model memory through retrieval enhancement holds significant applications in various domains (Sharma et al., 2021; He et al., 2023; Chen et al., 2022b), e.g., commonsense generation (Wang et al., 2021; Fan et al., 2020), dialogue systems (Song et al., 2016; Zhang et al., 2022; Zhu et al., 2019; Chen et al., 2022c; Parvez et al., 2023), summarization (An et al., 2021), and fact verification (Thorne et al., 2018).

Retrieval-enhanced machine learning methods could be mainly divided into two categories: (i) The first one is retrieval-only: the retrieval model acts as an auxiliary tool to provide query-related responses to the prediction model. Note that the majority of current fact verification models primarily belong to this category. (ii) The second one is retrieval with feedback: the retrieval model is further trained based on the feedback provided by the prediction model. Very recently, Hu et al. (2023) proposed to utilize the loss value of the prediction model as a signal for optimizing the retrieval model. However, this method may not be suitable for large datasets because it relies on all provided sentences to compute the loss. Additionally, without access to ground-truth evidence for the prediction model, accurately capturing the utility of retrieved sentences may be challenging.

## A.2  The training process of the fine-grained retriever

In this section, we provide a detailed description of training process of the fine-grained retriever, given the K candidate sentences from the coarse retriever, unfolds as follows:

The fine-grained retriever selects sentences from the candidate set.

- The claim and K candidate sentences $[s_1, s_2, \ldots, s_K]$ are concatenated with a special token [CLS] at the beginning of each sentence and then provided as input to the fine-grained retriever.
- Subsequently, the hidden state of each [CLS] undergoes a transformation through a linear layer followed by a ReLU activation, yielding K sets of 2-dimensional vectors, for example, [[0.1, 0.9], [0.2, 0.8], ..., [0.7, 0.3]].
- Finally, Gumbel Softmax is applied to these K 2-dimensional classification vectors, producing a

sequence-length K vector like [1, 1, ..., 0], where 1 indicates the predicted evidence sentence.

The fine-grained retriever forwards the selected sentence information to the verifier.

- The sequence-length K vector is adaptively extended to match the lengths of the K candidate sentences, aligning with the respective sentence lengths. For instance, $[len(s_1) * 1, len(s_2) * 1, \ldots, len(s_K) * 0]$ represents the selected vectors, which serve as the input_mask for the BERT tokenizer. Here, $len(s_1) * 1$ signifies the repetition of 1 for the length of $s_1$.

- The selected vectors $[len(s_1) * 1, len(s_2) * 1, \ldots, len(s_K) * 0]$ from the retriever, along with the K discrete sentences $[s_1, s_2, \ldots, s_K]$, are provided as input_mask and input_ids, respectively, to the verifier using the BERT tokenizer.

- For the ground-truth evidence, the input_ids corresponds to the true discrete evidence, while the input_mask are represented as the unit vector.

- Subsequently, the utility divergence between the verifier's judgments based on ground-truth evidence and those derived from retriever-provided sentences is computed, all in the context of predicting the claim label.

The verifier back-propagates the loss gradient to the fine-grained retriever.

- Following the computation of the utility divergence, gradients are back-propagated to the selected vectors, e.g., $[len(s_1) * 1, len(s_2) * 1, \ldots, len(s_K) * 0]$.

- As the Gumbel Softmax function is differentiable, gradients permeate through the selected vectors, extending back to the retriever.

### A.3 Reproducibility

In this section, we introduce our experimental details. The source code and trained models will be made publicly available upon publication to improve the reproducibility of results.

#### A.3.1 Experimental settings

For document retrieval, we adopt the entity linking approach (Hanselowski et al., 2018) to retrieve relevant documents, following the methodology of Hanselowski et al. (2018). On average, four documents are retrieved for each claim. Following the traditional entity linking pipeline approach, there are three steps for document retrieval. First, the claims are parsed using AllenNLP (Gardner et al., 2018) to extract multiple entities. Secondly, the

MediaWiki API[2] is utilized to search for page titles that correspond to the identified entities. Lastly, the retrieval results are filtered using the entity coverage limitation to select the appropriate documents.

For coarse retrieval, we follow the approach described in (Liu et al., 2020), where we utilize the base version of the BERT model (Devlin et al., 2019) from Hugging Face[3] to implement our coarse-grained retrieval model. We use the "[CLS]" hidden state to represent claim and sentence pairs. Training samples are constructed by using both ground-truth evidence and non-ground-truth sentences from document retrieval. The pairwise training method is employed to train the retrieval ranking model. The max length of the input is set to 130. We use the Adam optimizer with a learning rate of 5e-5 and a warm-up proportion of 0.1.

For fine-grained retrieval, it contains the fine-grained retrieval model $R_\theta$ and the verifier. Specifically, $R_\theta$ concatenates the claim and the retrieved candidate sentences and feeds them into the BERT model. The maximum length is set to 512, and the sentences are separated using the special token "[CLS]". The hidden vector corresponding to the "[CLS]" token is then used for evidence classification. The learning rate for the AdamW (Loshchilov and Hutter, 2019) optimizer is set to 2e-5. In the verifier model, the input consists of the concatenated sequence of the claim and the ground-truth evidence. The batch size is set to 5, and the accumulate step is also set to 5. The learning rate for the AdamW optimizer is set to 3e-5. The hyperparameters of $\alpha$ and $\beta$ are set to 1 and 1, respectively.

Furthermore, all models are implemented using the PyTorch framework. For online evidence evaluation, only the initial five sentences of predicted evidence provided by the candidate system are utilized for scoring. In order to adhere to the specifications of the online evaluation, the baselines and our FER select the five sentences as the evidence.

#### A.3.2 Evaluation metric

For evaluation on the FEVER benchmark dataset[4], we use the official evaluation metrics, i.e., **P**recision (P), **R**ecall (R), and **F1** for evidence retrieval; **FEVER** score and **L**abel **A**ccuracy (LA) for claim verification. Following the official evaluation (Thorne et al., 2018), we compute P@5, R@5 and

[2] https://www.mediawiki.org/wiki/API:Main_pagel
[3] https://huggingface.co/
[4] https://fever.ai/dataset/fever.html

F1@5. FEVER[5] evaluates accuracy based on the condition that the predicted evidence fully covers the ground-truth evidence. LA assesses the accuracy of the claim label prediction without taking into account the validity of the retrieved evidence.

### A.3.3 Evidence retrieval baselines

In our work, we consider several advanced evidence retrieval baselines.

- **TF-IDF** (Thorne et al., 2018) is a traditional sparse retrieval model that combines bigram hashing and TF-IDF matching to return relevant documents.
- **ColumbiaNLP** (Chakrabarty et al., 2018) initially utilizes TF-IDF to rank the candidate sentences and subsequently selects the top 5 sentences with the highest relevance. To mitigate the presence of noisy data, ELMo embeddings (Sarzynska-Wawer et al., 2021) are employed to convert the claims and sentences into vectors. Subsequently, the top-3 sentences with the highest cosine similarity are extracted as the final retrieval results.
- **UKP-Athene** (Hanselowski et al., 2018) introduces a sentence ranking model utilizing the Enhanced Sequential Inference Model (ESIM) (Hanselowski et al., 2018). This model takes a claim and a sentence as input, and the predicted ranking score is obtained by passing the last hidden layer of the ESIM through a single neuron.
- **GEAR** (Zhou et al., 2019) enhances the UKP-Athene model by introducing a threshold. Sentences with relevance scores higher than the threshold (set to 0.001) are filtered and considered as retrieval results.
- **Kernel Graph Attention Network (KGAT)** (Liu et al., 2020) utilizes both ESIM and BERT (Devlin et al., 2019) to construct evidence retrieval models, which are trained in a pairwise manner.
- **DREAM** (Zhong et al., 2020) employs the contextual representation models XLNet (Yang et al., 2019) and RoBERTa (Liu et al., 2019) to assess the relevance of a claim to each candidate evidence.
- **NSMN** (Nie et al., 2019a) employs a direct traversal approach, where all sentences are compared with the claim to calculate relevance scores. Sentences with relevance scores higher than the threshold (set to 0.5) are considered as evidence.

[5]https://codalab.lisn.upsaclay.fr/competitions/7308

Table 4: Performance of different claim verification models on the development set using evidence from the original paper vs. evidence retrieved by FER.

| Model | LA | FEVER |
|---|---|---|
| BERT Concat (Zhou et al., 2019) | 73.67 | 68.89 |
| BERT Concat + FER | 76.31 | 69.90 |
| BERT Pair (Zhou et al., 2019) | 73.30 | 68.90 |
| BERT Pair + FER | 76.34 | 69.35 |
| GEAR (Zhou et al., 2019) | 74.84 | 70.69 |
| GEAR + FER | 76.24 | 72.17 |
| GAT (Liu et al., 2020) | 76.13 | 71.04 |
| GAT+FER | 77.75 | 71.27 |
| KGAT(BERT Base) (Liu et al., 2020) | 78.02 | 75.88 |
| KGAT+FER | 79.02 | 76.59 |

Table 5: Evidence retrieval performance with feedback from different verifiers on the development set.

| Verifier | P | R | F1 |
|---|---|---|---|
| GAT | 82.45 | 85.65 | 84.02 |
| KGAT | 88.57 | 81.82 | 85.06 |

- **Deep Q-learning Network (DQN)** (Wan et al., 2021) utilizes the RoBERTa for evidence representation and applies the deep Q-learning network to select precise evidence.
- **GERE** (Chen et al., 2022a) employs the new paradigm of generative retrieval to generate the relevant evidence identifiers.
- **Stammbach** (Stammbach, 2021) employs token-based single-document candidate sentence classification to retrieve evidence, which uses RoBERTa (Liu et al., 2019) and BigBird (Zaheer et al., 2020) to encode candidate sentences. To ensure a fair comparison, we adopt the results based on RoBERTa as our baseline.

### A.3.4 Veracity prediction

In our work, we also leverage several advanced claim verification models for verification based on the retrieved evidence by FER.

- **BERT-pair and BERT concat** (Zhou et al., 2019) consider claim-evidence pairs individually, or stitch all evidence and the claim together to predict the claim label.
- **GEAR** (Zhou et al., 2019) considers the influence between evidence using a graphical attention network and aggregates all evidence through the attention layer.
- **KGAT** (Liu et al., 2020) introduces a graph attention network to measure the importance of evidence nodes and the propagation of evidence among them through node and edge kernels.

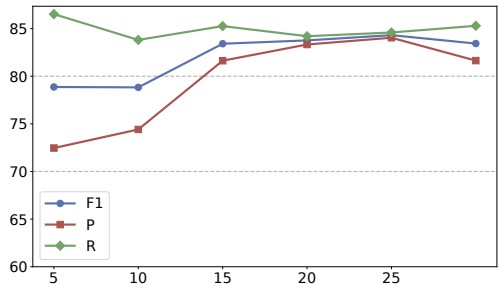

Figure 3: Evidence retrieval performance with different $K$ on the development set.

## A.4 Experimental results

### A.4.1 Performance on the development set.

As shown in Table 4, we report the performance of different claim verification models on the development set using the evidence retrieved by FER. As shown in Table 5, we report the evidence retrieval performance of our method with feedback from different verifiers on the development set. As shown in Figure 3, we report the effect of the size $K$ on the final evidence retrieval performance on the development set.

### A.4.2 Case study

Table 6 presents two illustrative examples from the FEVER development set. We can observe that: (i) The sentences provided by the coarse retrieval method, even though they are ranked highly, do not always offer useful information for claim verification, specifically in terms of providing ground-truth evidence. For example, in the second instance, out of the top-5 ranked sentences, only two of them actually consist of ground-truth evidence. (ii) Through fine-grained retrieval, we can effectively identify ground-truth evidence that might have initially been ranked low, such as the 14-th sentence in the first instance and the 11-th sentence in the second instance. (iii) In the case of fine–grained retrieval, not all ground-truth evidence are successfully identified. For example, in the second instance, the 8-th sentence is not identified. However, as observed in the given example, only the 8-th sentence is not retrieved, which does not significantly affect the judgment of the claims. Nonetheless, the issue of the low recall rate necessitates further resolution in future research.

Table 6: Two examples from the FEVER development set using our FER method. For a given claim, the coarse retrieval process returns the top-25 sentences. The ground-truth evidence is highlighted in red. The final evidence identified through our fine-grained retrieval process is {1, 14}, and {1, 2, 4, 5, 6, 7, 11}, respectively.

**Claim:** The Love Club EP is the debut extended play of singer Ella Marija Lani Yelich-O'Connor.

**Ground-truth evidence:**
1. The Love Club EP is the debut extended play EP by New Zealand singer Lorde · · ·
2. Ella Marija Lani Yelich O'Connor born 7 November 1996, better known by her stage name Lorde · · ·

**Label:** SUPPORT

**Coarse Retrieval Process:**
1. The Love Club EP is the debut extended play EP by New Zealand singer Lorde.
2. The Love Club is a song by New Zealand singer Lorde, taken from her debut extended play · · ·
3. An indie rock influenced electronica album, The Love Club EP · · ·
4. To promote The Love Club EP, Lorde performed during various concerts, and Royals was released · · ·
5. On 8 March 2013 the record was commercially released by Universal Music · · ·
6. Upon the release of the EP, the song was well received by music critics · · ·
· · ·
14. Ella Marija Lani Yelich-O'Connor born 7 November 1996 · · ·
· · ·

**Claim:** Pearl Jam is a type of dressing.

**Ground-truth evidence:**
1. Since its inception, the band's line up has comprised Eddie Vedder lead · · ·
2. Stephen Thomas Erlewine of AllMusic referred to Pearl Jam as the most popular American rock roll · · ·
3. The band's fifth member is drummer Matt Cameron also of Soundgarden, who has been · · ·
4. Pearl Jam is an American rock band formed in Seattle, Washington, in 1990.
5. Pearl Jam has outlasted and outsold many of its contemporaries from the alternative rock breakthrough · · ·
6. To date, the band has sold nearly 32million records in the United States and an · · ·

**Label:** REFUTES

**Coarse Retrieval Process:**
1. Pearl Jam is an American rock band formed in Seattle, Washington, in 1990.
2. Stephen Thomas Erlewine of AllMusic referred to Pearl Jam as the most popular · · ·
3. Pearl Jam sometimes referred to as The Avocado Album or simply Avocado is the eighth studio · · ·
4. One of the key bands in the grunge movement of the early 1990s, over the course of the band's career, its · · ·
5. Formed after the demise of Gossard and Ament's previous band, Mother Love Bone, Pearl Jam broke into the · · ·
6. Pearl Jam has outlasted and outsold many of its contemporaries from the alternative rock breakthrough · · ·
7. Since its inception, the band's line up has comprised Eddie Vedder lead · · ·
8. The band's fifth member is drummer Matt Cameron also of Soundgarden, who has been · · ·
· · ·
11. To date, the band has sold nearly 32million records in the United States and an · · ·
· · ·