# OpenReview forum: "From Relevance to Utility: Evidence Retrieval with Feedback for Fact Verification"
_EMNLP/2023/Conference — EMNLP 2023 Findings_

### Official Review · Reviewer_MWUn · 2023-07-31

**Soundness:** 3

**Excitement:**

4: Strong: This paper deepens the understanding of some phenomenon or lowers the barriers to an existing research direction.

**Paper Topic And Main Contributions:**

This paper focus on evidence retrieval for claim verification. They propose to use an additional loss function when training a retrieval model. This additional term comes from the claim verification model. They report improvement over retrieval and verification experiments.

**Questions For The Authors:**

Question A: Do you use a fixed candidate set size of 25 or is it the higher bound?
Question B: How different is you method with a simple method of changing a Threshold T to modify the classification decision? I was wondering if by adding a second term to the loss function you just balance something like a threshold. My clue to think about this possibility is your results in Table 1: We see that FER impose a balance between recall and precision between FER w/o L_cla and FER w/o L_uti.
Question C: You first use a coarse model and then train an extra fine-grained retrieval model. This clearly result in improving precision and decreasing recall (Table 1). How do you prove the final improvement comes from your extra loss and not just an addition point-wise model on top of your pair-wise results? Do you have the result of the pure coarse model too?
Question D:  L_cls is a point-wise loss function, why do you describe your ranking model as pair-wise in line 117? Do you mean your coarse retrieval is a pair-wise model and the fine-grained one is point-wise? Maybe you should clarify this confusion.

**Reasons To Accept:**

- Improving the evidence retrieval and consequently claim verification performance using their proposed new loss function.

**Reasons To Reject:**

It is not clear if the improvement in results stem from an additional point-wise model which is trained over the results of a coarse retrieval model or it is because of the proposed loss function.

**Reproducibility:**

3: Could reproduce the results with some difficulty. The settings of parameters are underspecified or subjectively determined; the training/evaluation data are not widely available.

**Reviewer Confidence:**

5: Positive that my evaluation is correct. I read the paper very carefully and I am very familiar with related work.

---

> ### Author Rebuttal · Authors · 2023-08-28
>
> We sincerely thank you for your invaluable feedback and insightful comments. Your time and dedication in evaluating our work are greatly appreciated. We would like to offer our response as follows.
>
> [Reject-1] Thanks for pointing out this. It is an important issue we should consider. Similar to the reply to the [Question-C].
>
> [Question-A] The candidate set size is fixed and set to 25, which is tuned on the validation set. In Figure 2 and 3, we also plot the effect of the size of candidate set (5, 10, 15, 20, 25 and 30) on the final evidence retrieval performance. We find that as the size is larger than 25, more noisy/unnecessary information is provided to the fine-grained retriever, potentially hurting precision.
>
> [Question-B] Thanks, this is an interesting perspective on the proposed loss.
>
> 1. A simple method of changing a Threshold has been used in existing works. For instance, the baseline model GEAR initially retrieves sentences with the highest 5 relevance scores. Subsequently, GEAR applies a relevance score filter using a designated threshold, which eliminates sentences with scores below that threshold. Consequently, the retrieved evidence set is equal to or less than 5. However, GEAR's performance in our paper's Table 1 and in the original GEAR paper's Table 4 is not particularly promising, even across various thresholds.
> 2. The hard threshold approach is heuristic, relying solely on relevance and disregarding downstream verification tasks, which results in a lack of feedback from them. In contrast, our method provides quantity flexibility and directly guides retrieval based on utility information derived from downstream verification tasks.
> 3. Your discovery on the balance between FER w/o L\_cla and FER w/o L\_uti is an interesting and profound issue. We think it might that different loss functions might emphasize distinct aspects. Traditional approach of using hard thresholding to filter relevance lacks flexibility. The L\_{cla} loss and L\_{uti} loss act like selecting individual appropriate soft thresholds for each independent claim, thus achieving an overall performance improvement.
> 4. Thank you again for this intriguing question and welcome further discussions. We will include the discussions in the revision.
>
> [Question-C] Thanks for your valuable comments.
>
> 1. For the coarse model used in our work, we follow the retrieval model in KGAT (Line 110). Consequently, the outcome of the pure coarse model aligns with KGAT's results in Table 1. Specifically, the pure coarse model (i.e., KGAT) exhibits rather low precision and F1 scores. Our approach achieves a remarkable 101.4% improvement over KGAT's F1 score on the test set.
> 2. In our work, we input the concatenated sequence of the claim and all the candidate sentences given by the coarse retriever, with a special token [CLS] added at the beginning of each sentence to indicate its presence, into the fine-grained retriever (Line 124-128). Then, we leverage the hidden state of each [CLS] to classify whether the corresponding candidate sentence serves as evidence or not. Therefore, our fine-grained retriever is not merely a point-wise training approach. The interaction information between all candidate sentences might also aid in classifying each sentence, which has been used in previous works [1, 2, 3].
> 3. The coarse retriever and fine-grained retriever are kept independent. As you suggested, if we only add the model of fine-grained retriever on top of the pair-wise results, it can be generally regarded as the variance of our model, i.e., FER w/o L\_uti (only with classification loss). As shown in Table 1, FER w/o L\_uti outperforms baselines, demonstrating the superiority of the addition model training strategy. However, FER w/o L\_uti underperforms FER w/o L\_cla and the full version FER in terms of precision, showing the importance of the novel utility divergency. Therefore, we think the performance improvement stems from changes in the training strategies as well as the introduction of new loss functions.
> 4. Thank you again for this insightful question. We will include the discussions in the revision. Should you still have any remaining concerns or queries regarding this matter, please don't hesitate to continue the discussion. Your input is highly valued.
>
> [Question-D] We are sorry for causing any confusion on your part. We further clarify the difference between the coarse retriever and the fine-grained retriever:
>
> 1. Yes, our coarse retriever is a pair-wise model. However, the fine-grained retriever is not merely point-wise, which is similar to the reply to the [Question-C].
> 2. The parameters of the coarse-retrieval and the fine-grained retrieval are not shared; they are kept separate and independent.
>    - The development of the coarse retriever involves pair-wise training, which remains grounded in relevance.
>    - Building upon the sentences identified by the coarse retriever, the fine-grained retriever focuses on extracting evidence that offers significant utility to the verifier.
> 3. Thanks again for your suggestions. We will provide clearer clarification of this differentiation in the revision.
>
> Reference:
>
> [1] Stammbach, D. (2021). Evidence selection as a token-level prediction task. In Proceedings of the Fourth Workshop on Fact Extraction and VERification (FEVER) (pp. 14-20). Association for Computational Linguistics.
>
> [2] David Wadden, Kyle Lo, Lucy Lu Wang, Arman Cohan, Iz Beltagy, and Hannaneh Hajishirzi. 2022. MultiVerS: Improving scientific claim verification with weak supervision and full-document context. In Findings of the Association for Computational Linguistics: NAACL 2022, pages 61–76, Seattle, United States. Association for Computational Linguistics.
>
> [3] Hu, X., Hong, Z., Guo, Z., Wen, L., & Yu, P. (2023, July). Read it Twice: Towards Faithfully Interpretable Fact Verification by Revisiting Evidence. In Proceedings of the 46th International ACM SIGIR Conference on Research and Development in Information Retrieval (pp. 2319-2323).

---

### Official Review · Reviewer_uHs1 · 2023-08-05

**Soundness:** 4

**Excitement:**

2: Mediocre: This paper makes marginal contributions (vs non-contemporaneous work), so I would rather not see it in the conference.

**Paper Topic And Main Contributions:**

This paper proposed the feedback-based evidence retriever (FER) that optimizes the evidence retrieval process by incorporating feedback from the claim verifier. The retriever is trained using both the direct supervision from the gold evidence and the indirect supervision from the final task loss. Experimental results on FEVER dataset (Thorne et al., 2018) demonstrated a 23.7% F1 performance gain over the SOTA baseline in terms of evidence sentence retrieval.

**Questions For The Authors:**

- I got confused when comparing the numbers in Table 1 and Table 2. In Table 1, FER outperforms all other baselines by a large margin. For instance, FER achieves 78.84 F1 score on the test set of FEVER while KGAT (Liu et al., 2020) only achieves 39.14 F1 score. However, in Table 2, the improvement on downstream performance (FEVER score) over KGAT is quite marginal: KGAT (69.40) vs. KGAT+FER (69.61). Do you have an explanation for this?

**Reasons To Accept:**

This paper utilized the idea of using downstream task loss to improve the intermediate component, here, the fine-grained retriever, and achieved strong retrieval results on FEVER dataset.

**Reasons To Reject:**

- One of the main weakness of this paper is the lack of excitement. The idea of using downstream task loss to improve the intermediate component is not new and has been explored by many prior work. For example, Choi et al. 2016 used QA loss as indirect supervision and using RL to improve the sentence retriever; Lewis et al., 2020 used the two different decoder loss to jointly train the retriever and the decoder. This work is an instantiation of such techniques on FEVER.
- I got confused by the equation at line 165. Here, if I understand it correctly, $R_\theta(c, S)$ represents the fine-grained retriever and it returns a "hard" sentences selection from $S$. If it's a "hard" selection, how is the loss gradient passed to the fine-grained retriever? I also got confused by the results in Table 1 and Table 2. See the following question section for details.

Reference:
- Choi, Eunsol, et al. "Coarse-to-fine question answering for long documents." Proceedings of the 55th Annual Meeting of the Association for Computational Linguistics (Volume 1: Long Papers). 2017.
- Lewis, Patrick, et al. "Retrieval-augmented generation for knowledge-intensive nlp tasks." Advances in Neural Information Processing Systems 33 (2020): 9459-9474.

**Reproducibility:**

3: Could reproduce the results with some difficulty. The settings of parameters are underspecified or subjectively determined; the training/evaluation data are not widely available.

**Reviewer Confidence:**

3: Pretty sure, but there's a chance I missed something. Although I have a good feel for this area in general, I did not carefully check the paper's details, e.g., the math, experimental design, or novelty.

---

> ### Author Rebuttal · Authors · 2023-08-28
>
> We sincerely thank you for your invaluable feedback and insightful comments. Your time and dedication in evaluating our work are greatly appreciated. We would like to offer our response as follows.
>
> [Reject-1] Yes, many studies have delved into leveraging the loss of the downstream task to enhance the intermediary component. These works primarily differ in two crucial aspects: (1) the design of feedback provided by the downstream task, and (2) the utilization of this feedback in the intermediate component.
>
> 1. For feedback design: Previous studies, e.g., Choi et al. (2016) and Lewis et al. (2020), directly leveraged downstream task performance using retrieval results as feedback. Nonetheless, this strategy heavily depends on the prediction accuracy of downstream task models. In contrast, we evaluate the divergence in utility between verifier assessments grounded in ground-truth evidence and those derived from retrieval outcomes. This approach mitigates the extent of reliance on the downstream task models' precision.
> 2. For feedback use: In prior studies, such as Choi et al. (2016) and Lewis et al. (2020), employed reinforcement learning techniques for parameter updates. In contrast, we utilize utility divergence as the retriever's training loss directly. While our baseline model, DQN, used reinforcement learning, our approach significantly outperforms it, achieving a 26.0% higher F1 score on the test set. The reason may be that reinforcement learning involves intricate reward function design and optimal policy formulation, which can result in unintended agent behaviors if not done well. Our direct optimization approach could eliminate the need for intricate reward design and policy tuning.
> 3. We contribute two major improvements as described above, rather than merely instantiating existing techniques on FEVER.
> 4. In the future works, we will explore alternative ways for feedback design and model training. Besides, we will incorporate a discussion on these two aspects within our revision.
>
> [Reject-2] We are sorry for any confusion caused. To clarify, the retriever employs a soft selection approach for sentences, as opposed to a hard one. In other words, the retriever's chosen sentences are used as vector-based input for the verification model, rather than as discrete sentences.
>
> The training process of the fine-grained retriever, given the K candidate sentences from the coarse retriever, unfolds as follows:
>
> 1. The fine-grained retriever selects sentences from the candidate set.
>    - The claim and K candidate sentences [s\_1, s\_2, …, s\_K] are concatenated with a special token [CLS] at the beginning of each sentence, and then provided as input to the fine-grained retriever.
>    - Subsequently, the hidden state of each [CLS] undergoes a transformation through a linear layer followed by a ReLU activation, yielding K sets of 2-dimensional vectors, for example, [[0.1, 0.9], [0.2, 0.8], ..., [0.7, 0.3]].
>    - Finally, Gumbel Softmax is applied to these K 2-dimensional classification vectors, producing a sequence-length K vector like [1, 1, ..., 0], where 1 indicates the predicted evidence sentence.
> 2. The fine-grained retriever forwards the selected sentence information to the verifier.
>    - The sequence-length K vector is adaptively extended to match the lengths of the K candidate sentences, aligning with the respective sentence lengths. For instance, [len(s\_1)\*1, len(s\_2)\*1, …, len(s\_K)\*0] represents the selected vectors, which serve as the input\_mask for the BERT tokenizer. Here, len(s\_1)\*1 signifies the repetition of 1 for the length of s\_1.
>    - The selected vectors [len(s\_1)\*1, len(s\_2)\*1, …, len(s\_K)\*0] from the retriever, along with the K discrete sentences [s\_1, s\_2, …, s\_K], are provided as input\_mask and input\_ids, respectively, to the verifier using the BERT tokenizer.
>    - For the ground-truth evidence, the input\_ids corresponds to the true discrete evidence, while the input\_mask are represented as the unit vector.
>    - Subsequently, the utility divergence between the verifier's judgments based on ground-truth evidence and those derived from retriever-provided sentences is computed, all in the context of predicting the claim label.
> 3. The verifier back-propagates the loss gradient to the fine-grained retriever.
>    - Following the computation of the utility divergence, gradients are back-propagated to the selected vectors, e.g., [len(s\_1)\*1, len(s\_2)\*1, …, len(s\_K)\*0].
>    - As the Gumbel Softmax function is differentiable, gradients permeate through the selected vectors, extending back to the retriever.
>
> 4. We will include detailed implementation information in our revision and share the source code publicly after publication. For an explanation of the results in Table 1 and Table 2, please refer to the response to [Question-1].
>
> [Question-1] We are sorry for causing any confusion about the experimental results on your part. We further clarify the difference between Table 1 and Table 2 as follows:
>
> 1. Table 1 shows the evidence retrieval results on both the dev set and the online test set, which holds our primary focus in this study.
>    - We have two types of baselines: (1) 5-sentence evidence baseline: For TF-IDF, ColumbiaNLP, UKP-Athene, NSMN, KGAT and DREAM, they directly select the top 5 sentences as evidence for evaluation; (2) precise evidence baseline: For DQN, GEAR, GERE and Stammbach, they also emphasize the significance of achieving dynamic predictions for relevant sentences that suit different claims, i.e., a precise set of relevant sentences. This way often results in a predicted count that is potentially equal to or less than 5, like our results.
>    - In general, precise evidence baselines and our method outperform 5-sentence evidence baselines. Our approach outperforms GERE on the test set, resulting in a 23.7% increase in the F1 score.
> 2. Table 2 shows the claim verification results. We choose several advanced (off-the-shelf) claim verification models, provided with the retrieved evidence obtained by our method and baselines respectively, for further comparison.
>    - For these claim verification models, they have separate processes for their retrievers and verification models. Firstly, they leverage traditional retrieval methods, such as BM25, BERT based sentence retrieval and other retrieval methods based on relevance, to obtain relevant sentences; Subsequently, they usually train the verification models using the ground truth data with added noise, rather than using the retriever's retrieval results.
>    - Hence, some verification models’s structure designed for noisy training data might not be very sensitive to retrieved evidence, even though the retrieval performance we've achieved has seen significant improvement (Table 1). For example, KGAT employs a graph attention network to explicitly model the mechanism of importance between evidence with respect to claims. This approach could potentially be robust to noisy evidence on claim judgments. Therefore, the difference between relying on evidence retrieved via previous methods and our approach might not be considerably large.
>    - The FEVER score's definition varies from the LA score. While the LA score assesses claim verification accuracy independently of retrieved evidence validity, different degrees of LA metric improvement are noticeable. The FEVER metric requires both accurate claim classification and inclusion of all ground-truth evidence. Thus, cases with modest LA score improvements tend to result in constrained improvements in FEVER scores.
> 3. Similar findings are also present in other studies on evidence retrieval [1], indicating large improvements on evidence retrieval and marginal improvements on the downstream claim verification given by the retrieved evidence.
> 4. Currently, our focus is solely on evidence retrieval, with separate optimization of retrieval and verification models. We intend to investigate methods for jointly optimizing both models within a single objective. In this manner, we expect to see improved FEVER and LA scores.
> 5. Thank you for this insightful question. We will polish the related content in our revision.
>
> Reference:
>
> [1] Chen, J., Zhang, R., Guo, J., Fan, Y., & Cheng, X. (2022, July). GERE: Generative evidence retrieval for fact verification. In Proceedings of the 45th International ACM SIGIR Conference on Research and Development in Information Retrieval (pp. 2184-2189).

---

### Official Review · Reviewer_tnjn · 2023-08-11

**Soundness:** 3

**Excitement:**

3: Ambivalent: It has merits (e.g., it reports state-of-the-art results, the idea is nice), but there are key weaknesses (e.g., it describes incremental work), and it can significantly benefit from another round of revision. However, I won't object to accepting it if my co-reviewers champion it.

**Paper Topic And Main Contributions:**

This paper focuses on evidence retrieval in fact verification tasks and argues that the sentences retrieved by existing methods may not align well with a verification model based on the retrieved evidence. The proposed method involves two phases: 1) coarse retrieval, which retrieves a smaller candidate set, and 2) fine-grained retrieval, which selects evidence for verification.

Specifically, the fine-grained evidence retriever is trained using a combination of classification loss and divergence utility. Its goal is to align the retrieval model with the verification model by incorporating the verdict results from the verifier, both from ground-truth evidence and from retrieved evidence.

Experiments conducted on FEVER demonstrate that the proposed method significantly improves the F1 performance of evidence retrieval.

**Questions For The Authors:**

1) Could you provide additional details on how the fine-grained retriever is optimized with feedback? It seems that the retriever predicts probabilities for sentences, which selected sentences are then used as input for the verification model. How is the gradient computed based on discrete sentence input?

2) The paper mentions conducting an online evaluation with the top 5 sentences as the evidence set, but the results are not reported. Could you please share the results of this evaluation?

3) The input of the verification model consists of retrieved sentences. Since the evidence set contains both ground-truth evidence and noise/unnecessary evidence, can the divergence signal force the fine-grained retriever to select the ground-truth evidence?

**Reasons To Accept:**

1) This work proposes a novel method to align evidence retrieval with verifier based on divergence of verdict prediction result of  ground-evidence and retrieved evidence as feedback signal.

2) The method is simple and effective. Experiments on FEVER show that FER improves the F1 score of evidence retrieval by a large margin.

**Reasons To Reject:**

1) The explanation of fine-grained retrieval is unclear and could be more concise. The process of constructing the final evidence set is not detailed enough, only mentioning online evaluation in lines 715-716 (Top5 sentences).

2) Lines 222-224 suggest that FER aims for more precise and concise evidence sets, potentially containing less than 5 sentences, resulting in higher precision and lower recall. The comparison with baselines using top 5 sentences for F1 score computation may not be completely fair.

**Reproducibility:**

4: Could mostly reproduce the results, but there may be some variation because of sample variance or minor variations in their interpretation of the protocol or method.

**Reviewer Confidence:**

4: Quite sure. I tried to check the important points carefully. It's unlikely, though conceivable, that I missed something that should affect my ratings.

---

> ### Author Rebuttal · Authors · 2023-08-28
>
> We sincerely thank you for your invaluable feedback and insightful comments. Your time and dedication in evaluating our work are greatly appreciated. We would like to offer our response as follows.
>
> [Reject 1]: Thanks for pointing out this, and we will provide detailed descriptions in our revision.
>
> 1. During training, for the fine-grained retriever, we take advantage of fine-tuning the pre-trained architecture (i.e., we use BERT weights) via the evidence classification loss L\_{cla} and utility divergency L\_{uti}.
> 2. During inference, given a test claim, we leverage the optimized retriever to construct the final evidence set as follows.
>    * Firstly, we obtain candidate sentences via the coarse retriever.
>    * Then, we input the concatenated sequence of the test claim and all the candidate sentences, with a special token [CLS] added at the beginning of each sentence to indicate its presence, into the fine-grained retriever (Line 124-128).
>    * Finally, we leverage the hidden state of each [CLS] to classify whether the corresponding candidate sentence serves as evidence or not. This classification is accomplished using linear layers and ReLU activation functions (Line 138-142). As a result, we are able to derive the final evidence set.
>    * This way has been used in previous fact checking [1,2] and other areas, such as extractive summarization [3].
> 3. For online evaluation, the FEVER online platform stipulates that the count of evidence must not exceed 5. Regarding the evidence extracted through the above process,
>    - If the count surpasses 5, we will exclusively consider the top 5 evidence for evaluation.
>    - Conversely, if the count falls below 5, we will directly incorporate them in their entirety.
>    - To ensure a fair comparison, we uniformly employ 5 evidence as provided by baselines.
>    - This evaluation way has been leveraged by many previous works, such as our baseline method: DQN, GERE and Stammbach.
>
> [Reject 2]: Thanks. This is indeed an essential aspect that we have taken into careful consideration.
>
> 1. The baselines we have employed can be broadly categorized into two distinct types:
>    - 5-sentence evidence baseline: TF-IDF, ColumbiaNLP, UKP-Athene, NSMN, KGAT and DREAM follow a process of ranking candidate sentences and subsequently utilize the top 5 ranked sentences as evidence.
>    - Precise evidence baseline: In the cases of DQN, GEAR, GERE and Stammbach, the authors also emphasize the significance of achieving dynamic predictions for relevant sentences that suit different claims, i.e., a precise set of relevant sentences. In this way, the final size of the retrieved evidence set is potentially equal to or less than 5, like our results.
> 2. It is worth noting that the FEVER dataset holds an average ground-truth evidence count of 1.8.
>    - For the 5-sentence evidence baselines, a direct selection of the top 5 sentences could introduce more noise, thereby potentially leading to reduced precision.
>    - In comparison with the 5-sentence evidence baselines, the precise evidence baselines as well as our method demonstrate an enhanced capability in accurately extracting evidence. Specifically, our approach surpasses GERE on the test set, resulting in a 46.1% increase in the precision score.
> 3. Therefore, following precise evidence baselines, we believe our approach to comparison is fair. We will include the related discussions in our revision.
>
> [Question-1] Thanks for pointing out this and we apologize for any confusion caused. To clarify, the sentences selected by the retriever are utilized as input to the verification model in vector form, rather than discrete sentences.
>
> The training process of the fine-grained retriever, given the K candidate sentences from the coarse retriever, unfolds as follows:
>
> 1. The fine-grained retriever selects sentences from the candidate set.
>    - The claim and K candidate sentences [s\_1, s\_2, …, s\_K] are concatenated with a special token [CLS] at the beginning of each sentence and then provided as input to the fine-grained retriever.
>    - Subsequently, the hidden state of each [CLS] undergoes a transformation through a linear layer followed by a ReLU activation, yielding K sets of 2-dimensional vectors, for example, [[0.1, 0.9], [0.2, 0.8], ..., [0.7, 0.3]].
>    - Finally, Gumbel Softmax is applied to these K 2-dimensional classification vectors, producing a sequence-length K vector like [1, 1, ..., 0], where 1 indicates the predicted evidence sentence.
> 2. The fine-grained retriever forwards the selected sentence information to the verifier.
>    - The sequence-length K vector is adaptively extended to match the lengths of the K candidate sentences, aligning with the respective sentence lengths. For instance, [len(s\_1)\*1, len(s\_2)\*1, …, len(s\_K)\*0] represents the selected vectors, which serve as the input\_mask for the BERT tokenizer. Here, len(s\_1)\*1 signifies the repetition of 1 for the length of s\_1.
>    - The selected vectors [len(s\_1)\*1, len(s\_2)\*1, …, len(s\_K)\*0] from the retriever, along with the K discrete sentences [s\_1, s\_2, …, s\_K], are provided as input\_mask and input\_ids, respectively, to the verifier using the BERT tokenizer.
>    - For the ground-truth evidence, the input\_ids corresponds to the true discrete evidence, while the input\_mask are represented as the unit vector.
>    - Subsequently, the utility divergence between the verifier's judgments based on ground-truth evidence and those derived from retriever-provided sentences is computed, all in the context of predicting the claim label.
> 3. The verifier back-propagates the loss gradient to the fine-grained retriever.
>    - Following the computation of the utility divergence, gradients are back-propagated to the selected vectors, e.g., [len(s\_1)\*1, len(s\_2)\*1, …, len(s\_K)\*0].
>    - As the Gumbel Softmax function is differentiable, gradients permeate through the selected vectors, extending back to the retriever.
> 4. We will incorporate the implementation specifics in our revision and make the source code publicly available upon publication. Thanks again for pointing out this.
>
> [Question-2] Similar to the response to [Reject 1] and [Reject 2], the results in Table 1 are derived from the top 5 sentences forming the evidence set. The precise evidence baselines, including DQN, GEAR, GERE, and Stammbach, utilize the same comparative approach. We will ensure to clarify this configuration in our revision.
>
> [Question-3] Thanks for your question.
>
> 1. The ground-truth evidence set and the retrieved sentences from the fine-grained retriever, are input separately into the verifier, generating two distinct probability distributions for feedback. Although the retrieved sentences may include some ground-truth evidence and noise evidence, the divergence signal can effectively guide the fine-grained retriever to focus solely on predicting the ground-truth evidence set.
> 2. Such an approach, founded on divergence between ground-truth and predicted information, finds its application in various domains, including instances like face recognition [4] and authorship attribution [5].
> 3. In Table 1, FER w/o L\_cla (only with utility divergence loss) outperforms GERE by 19% in terms of F1 on the test set. This underscores the pivotal role that feedback from the verifier plays in enhancing the retriever's ability to identify precise evidence.
> 4. We will include this discussion in our revision.
>
> Reference:
>
> [1] David Wadden, Kyle Lo, Lucy Lu Wang, Arman Cohan, Iz Beltagy, and Hannaneh Hajishirzi. 2022. MultiVerS: Improving scientific claim verification with weak supervision and full-document context. In Findings of the Association for Computational Linguistics: NAACL 2022, pages 61–76, Seattle, United States. Association for Computational Linguistics.
>
> [2] Hu, X., Hong, Z., Guo, Z., Wen, L., & Yu, P. (2023, July). Read it Twice: Towards Faithfully Interpretable Fact Verification by Revisiting Evidence. In Proceedings of the 46th International ACM SIGIR Conference on Research and Development in Information Retrieval (pp. 2319-2323).
>
> [3] Yang Liu and Mirella Lapata. 2019. Text Summarization with Pretrained Encoders. In Proceedings of the 2019 Conference on Empirical Methods in Natural Language Processing and the 9th International Joint Conference on Natural Language Processing (EMNLP-IJCNLP), pages 3730–3740, Hong Kong, China. Association for Computational Linguistics.
>
> [4] Wu, H., Xu, Z., Zhang, J., Yan, W., & Ma, X. (2017, October). Face recognition based on convolution siamese networks. In 2017 10th International Congress on Image and Signal Processing, BioMedical Engineering and Informatics (CISP-BMEI) (pp. 1-5). IEEE.
>
> [5] Fabien, M., Villatoro-Tello, E., Motlicek, P., & Parida, S. (2020, December). BertAA: BERT fine-tuning for Authorship Attribution. In Proceedings of the 17th International Conference on Natural Language Processing (ICON) (pp. 127-137).

---

### Meta-Review · Area_Chair_AY9S · 2023-09-18

**Recommendation:** 4

**Metareview:**

This paper aims to optimize the evidence retrieval process by utilizing feedback from the claim verifier. The retriever is trained on both ground-truth evidence and retrieved evidence. Fact verification is an important task and the proposed approach is intuitive and effective. The experiments showed good improvement on FEVER. Reviewers raised issues about unclear presentation in some parts of the paper, missing references, and weak ablation studies. One reviewer had a concern about lack of excitement. Overall, I think this paper presents a solid approach to an important task with some convincing results, especially considering it is a short paper. I hope the reviewers’ comments on the weaknesses will help improve the quality of the next version of the paper.

---

### Decision · Program_Chairs · 2023-10-07

**Decision:**

Accept-Findings

**Comment:**

This paper aims to optimize the evidence retrieval process by utilizing feedback from the claim verifier. The retriever is trained on both ground-truth evidence and retrieved evidence. Fact verification is an important task and the proposed approach is intuitive and effective. The experiments showed good improvement on FEVER. Reviewers raised issues about unclear presentation in some parts of the paper, missing references, and weak ablation studies. One reviewer had a concern about lack of excitement. Overall, I think this paper presents a solid approach to an important task with some convincing results, especially considering it is a short paper. I hope the reviewers’ comments on the weaknesses will help improve the quality of the next version of the paper.